# Air Pollution in a Nationally Representative Sample: Findings from the National Adult Tobacco Survey of Lao PDR

**DOI:** 10.3390/ijerph16183500

**Published:** 2019-09-19

**Authors:** Glorietta Hurd-Kundeti, Anne Berit Petersen, Khamphithoun Somsamouth, Pramil N. Singh

**Affiliations:** 1California Department of Public Health, 1615 Capitol Ave, Sacramento, CA 95814, USA; gahurd@gmail.com; 2Loma Linda University School of Nursing, 11262 Campus Street, Loma Linda, CA 92350, USA; abpetersen@llu.edu; 3Center for Health Research, Loma Linda University School of Public Health, 11234 Anderson St, Loma Linda, CA 92354, USA; 4Center for Information and Education on Health, Simuang Road, Vientiane Capital, Laos; kpsom1967@gmail.com; 5Transdisciplinary Tobacco Research Program, Loma Linda University Cancer Center, 11234 Anderson St, Loma Linda, CA 92354, USA

**Keywords:** Environmental tobacco smoke, air pollution, respiratory

## Abstract

In Southeast Asia, household air pollution (HAP) from solid fuel use is the leading cause of disability-adjusted life years (DALYs), a risk which is compounded by exposure to other sources of indoor and outdoor air pollution including secondhand tobacco smoke (SHS). The purpose of this study was to measure the individual and combined prevalence of exposure to household and community sources of air pollution in a national sample of adults in Lao PDR. We analyzed data from the 2012 National Adult Tobacco Survey (NATSL) of Lao PDR—a multi-stage stratified cluster sample of 9706 subjects from 2822 households located in all 17 provinces. Our findings indicate a high prevalence of exposure to household air pollution from cooking fires (78%) and SHS exposure in the home (74.5%). More than a third (32.8%) reported exposure to both inside the home. Exposure to outdoor sources of smoke from cooking, trash, and crop fires was substantial (30.1% to 56.0%). The aggregation of exposures from multiple sources of household air pollution raises the need for initiatives that establish programmatic linkages in the health, environmental, and agricultural sectors to provide a comprehensive strategy to reduce risk factors for respiratory disease in Lao PDR and the region.

## 1. Introduction

In 2016, 7 million premature deaths were attributed to ambient (outdoor) and household air pollution, with more than 90% of these premature deaths occurring in low- and middle-income countries (LMIC) [1,2]. While 4.2 million deaths are attributed to ambient air pollution, approximately 3.8 million deaths are attributable to household air pollution produced directly from the burning of biomass (solid) fuels (wood, crop waste, dung, and coal) for cooking and heating [3]. Globally, it is estimated that 3 billion people are reliant on biomass fuel, a practice which places them at risk for prolonged and extensive exposure to fine particulate matter (PM) (<2.5 μm in aerodynamic diameter (PM_2.5_)) carbon monoxide, and other health-damaging pollutants [2]. According to the World Health Organization (WHO), in poorly ventilated dwellings, it is not unusual for indoor exposure levels from biomass fuel to exceed acceptable fine PM levels by 100-fold, with women and young children experiencing the highest levels of exposure due to the amount of time spent inside the home [4]. 

In LMIC, household air pollution from solid fuel contributes to 12% of ambient air pollution globally. In South Asia and China, this estimate could be as high as 30% [5]. According to the Global Burden of Disease Report, household air pollution is the leading cause of disability-adjusted life years (DALYs) in Southeast Asia and the Western Pacific Region, and the third leading cause of DALYs globally [1]. 

When considering the health effects of high levels of exposure to smoke from solid fuels in the Western Pacific Region, it is important also to consider the potential for that risk to be compounded by other highly prevalent sources of household air pollution such as secondhand smoke from tobacco (SHS) [6]. In the Western Pacific region, nearly half of all adult men are current tobacco smokers [7]. The resultant toll on society is immense, with approximately 1 million cardiovascular disease deaths alone occurring each year from direct use of tobacco, while an additional 168,000 cardiovascular-related deaths per year are due to heart disease and stroke as a result of non-smokers exposure to SHS [2]. 

In the present study, we analyzed reports of exposure to SHS, and various types of household and ambient air pollution collected during the 2012 National Adult Tobacco Survey of Laos (2012 NATSL) [8]. The 2012 NATSL is the first and largest national adult (ages 15 years and older) tobacco survey conducted in Lao PDR and was completed through a collaboration between the Lao Statistics Bureau, Lao PDR Ministry of Health, National Institute of Public Health (Lao PDR), and Loma Linda University (Fogarty/NIH Asia Tobacco Control Leadership Training Program) [9]. In addition to a detailed section on use of all forms of smoked tobacco (Global Adult Tobacco Survey items) [10], detailed data on exposure to SHS from tobacco, household air pollution from cooking fires, and ambient air pollution from crop burning was collected using a set of picture cards of rural daily life in Laos. In our analysis of the sample of 9706 adults (age 15 years and older) enrolled, the specific aims are as follows to (1) identify the demographics of those households where members are exposed to SHS, (2) characterize prevalence of exposures to multiple personal (household) and community sources of air pollution (indoor cooking fires, crop burning, trash burning) with the potential to compound the already large burden of SHS exposure from tobacco. Demographic subgroups of the population, where exposures are particularly high, are identified, and implications discussed.

## 2. Methods

### 2.1. Study Population and Procedures

The 2012 National Adult Tobacco Survey of Laos (NATSL 2012) enrolled 9706 subjects ages 15 years and older who were present at home and consented to survey participation [11]. Using the 2010 census as a sampling frame, the 2012 NATSL sample was selected using a stratified, multi-stage cluster sample described in detail elsewhere [11,12]. Laos was stratified into 17 census-derived survey domains composed of 12 individual provinces and five groups of similar provinces. For the first stage of sampling, 25–26 primary sampling units (PSU) from each domain (i.e., villages or comparable urban unit) were selected to ensure 90% statistical power to estimate national prevalence within 2% accuracy. Each PSU was converted into number of households and number of enumeration areas (EA). An EA was selected from each PSU based on probability proportional to the EA size. A list of households in each selected EA served as the sampling frame for selection of households. EA’s that had more than 200 households were segmented, and only one segment was randomly selected for inclusion in the survey—the selection probability was then proportional to the final segment size. 

Survey teams consisted of four or five trained interviewers and enumerators who worked in each of the 17 census-derived regions described above. A total of 86 people were trained on the specific procedures for administering 2012 NATSL survey items and pictograms [13,14] (see Figure 1), along with two of the report authors, during a one-week session that preceded the data collection efforts. Detailed procedures for administering 2012 NATSL survey items and pictograms and evaluation of validity and reliability of survey items and pictograms for use in a national household survey have been presented elsewhere [14].

The sampling frame consisted of 9706 adults selected from households inclusive of all households from all 17 provinces that were private and either single or multiple members private and single-member households from all provinces. The survey did not include institutional households such as military barracks, prisons, hospitals, and residents of temples. The final analytic sample reduced to 9043 adults with complete data on respiratory exposures and disease.

Written informed consent was obtained from each subject, and the protocols for the 2012 NATSL national survey and its sub-studies were approved by the Institutional Review Board of Loma Linda University and Ethics review by the Ministry of Health of Lao PDR. An incentive equivalent to approximately US $0.50 was provided to each participant. This NIH sponsored project (project number 2R01TW005964-06) in Lao PDR is approved by IRB #5170182 by the Loma Linda University Institutional Review Board since 6/16/2017. 

### 2.2. Questionnaire

The content and design of the survey were developed in English by an international team of investigators. The final survey was conducted in the local language after the written survey items had been translated and back-translated to verify content, criteria and semantic equivalence by bilingual and monolingual experts using the methods described by Flaherty et al. [15].

During data collection, manual editing was performed by field supervisors from the National Institute of Statistics (Laos PDR). After the survey was returned, the data entry, verification, coding, and data cleaning was accomplished using the Census and Survey Processing System (CSPro version 5) software package (CSPro, Suitland, MD, USA) [16].

### 2.3. Statistical Analysis

The smoked (combustible) tobacco exposure variables used in the analysis, were created using a standardized coding method for Global Adult Tobacco Survey (GATS) items that classify subjects as daily smokers, less than daily smokers, and non-smokers—both currently and in the past.

Prevalence of exposure to indoor cooking fires and SHS and various sources of outdoor smoke in the national sample were reported for pertinent demographic variables including age, gender, village type, income, and education. To account for the stratified multi-stage cluster design, the variance for calculating 95% confidence intervals for measures of prevalence were computed using a Taylor series linearized method [14]. This method allowed for the computation of between-cluster variance estimators that accounted for the intra-cluster correlation among subjects within the same village. Point estimates were further adjusted by sample weights to account for different sampling fractions within each of the 17 domains. Statistical analyses were performed with SUDAAN (RTI international, Research Triangle Park, NC, USA) (software [17]).

## 3. Results

Overall, more than half of all adults in the national sample reported exposure to SHS in the home (74.5%, 95% CI = (70.9 to 77.8)) or indoor cooking fire smoke (78.0%, 95% CI = (74.8 to 80.8)) (Table 1). In Table 1, we provide the prevalence of exposure to individual sources of household and community air pollution by demographic variables. Participants’ in urban areas reported lower levels of exposure to all sources of air pollution. However, when analyzed by income, more than half of all adults were exposed to SHS indoors and cooking fire smoke (indoor and outdoor) across all income levels. Overall, the most common sources of exposure to air pollution occurred indoors and in the home. 

In Table 2, we report the overall prevalence of multiple sources of exposure, including SHS in the home, indoor cooking fires, outdoor cooking fires, trash burning, and crop burning. Overall, a third of all adults in the national sample were exposed to both SHS and cooking fires (indoors). Additionally, in Table 2, the prevalence of these multiple exposures to household and community sources of air pollution are presented by demographic variables. There was an unexpected trend of a higher prevalence of exposure to both SHS and indoor cooking fires in urban areas and among adults with higher levels of education. However, this contrast attenuated when adding other exposures, and it is noted that confidence intervals were wide.

## 4. Discussion

We examined the prevalence of multiple sources of household and community sources of air pollution in a national sample of 9,043 adults enrolled in the largest national survey of adult tobacco use conducted to-date in Lao PDR. Our major finding was that household air pollution from both cooking fires and tobacco, was highly prevalent, as nearly three-quarters of all households surveyed reported exposure to smoking in the home and exposure to indoor cooking fire smoke. When multiple sources of exposure were evaluated together, a third of all households reported household level exposure to both SHS from tobacco and smoke from indoor cooking fires. 

The high rates of combined exposure to multiple sources of indoor and outdoor smoke found in this study represent significant sources of health risks. For example, from this sample, we recently reported that for a set of environmental risk factors (smoked tobacco, environmental tobacco smoke in an enclosed workspace, indoor cooking fire, trash fires, other outdoor communal fires), each incremental exposure added a significant 47% increase in odds of tuberculosis [13]. Thus, for the Lao smoker who already experiences a significant 47% increase in odds of tuberculosis under our model, the addition of one to four additional pollutant exposures (environmental tobacco smoke in enclosed workspace, indoor cooking fire, trash fires, other outdoor communal fires) adds another 47% to 188% increase in odds of disease.

Another study from our group was conducted by Lopez et al. [18], and explored the effects of exposure to multiple pollutants on respiratory health among adult males in Lao PDR and found that combined exposure to tobacco, animal handling, cooking fires, and dust/dirt were associated with a 2-fold increase in odds of a positive COPD screen. While the present study included only adults, our findings reflect household level exposure, which in turn represent serious concerns for the infants and children living in households with multiple sources of household air pollution. A study conducted in Lao PDR that analyzed the effects of housing characteristics and activities on the respiratory health among women and children between ages 1 to 4 years old found that respiratory illness was associated with the location of fire, location of the cooking place, and smoking within the home [19]. Household air pollution exposure becomes an even more significant issue for adults, and children in particular, living in households located in the North of Lao PDR where indoor cooking fire exposure is prolonged due to its dual purpose of cooking and keeping the home warm [6,20]. 

The evidence for major health risks associated with household air pollution from biomass fuel is growing and continues to mirror many of the risks long associated with SHS exposure, including coronary heart disease, stroke, COPD, pneumonia, and lung cancer, and negative birth outcomes such as low birth weight, prematurity, and stillbirth, as well unique risks such as an increased risk of cataracts [6,21,22,23,24]. In the Western Pacific Region, research involving household air pollution has been conducted across varied settings including China [25,26], Singapore [27], Thailand [28], Philippines [29], Vietnam [30], Cambodia [31], and Lao PDR [18,19,32,33]. Studies in the Western Pacific Region have estimated use of biomass fuel indoors to be high, representing a high proportion of the disease burden, which only increases with the presence of smoked tobacco in the home [1,34,35,36]. One study, conducted in Lao PDR, also demonstrated that the health effects of household air pollution and its health implications arise from a complex framework of personal behaviors, community behaviors and the built environment [18]. Our findings from a large, representative national sample of Lao PDR add to the mounting evidence associated with implicating high levels of household air pollution. In Lao PDR, these risks are exacerbated by the urbanization underway within the country and surrounding nations such as China whose industries, due to fossil fuel emissions and automobile activity, are compounding the burden of outdoor air pollution [37]. 

While levels of more than 60 documented sources of household air pollution may vary from country to county, SHS consistently represents a major source of household air pollution [6]. It is well established that there are no safe levels of exposure to SHS [22]. Over the past three decades, while smoking has declined in high-income countries, the smoking prevalence has steadily increased in LMIC—where currently about 80% of the world’s 1.1 billion smokers live [38]. Based on global tobacco projections, exposure to SHS is increasingly compounding the profile of environmental health risks across many LMIC. 

A 2008 study, conducted to characterize SHS exposure among women and children across 31 countries, using air and hair nicotine exposure, found high levels of exposure in households with a smoker (median = 17 times higher, 0.18 µg/m^3^) compared with households without smokers (0.01 µg/m^3^) [39]. Overall, 82% of smokers indicated that they smoked inside the home, and around their children [39]. The highest levels of exposure were observed in samples from Asia, which was attributed primarily to a lack of voluntary smoking restrictions in households and around children [39,40]. However, globally, enormous progress has been made in tobacco control since adoption of the WHO Framework Convention on Tobacco Control in 2003, and the MPOWER tobacco control interventions continue to help countries implement and monitor measures to reduce the demand for tobacco and protect people from tobacco smoke [41].

In addition to the smoke from cooking fires and tobacco, reports from rural regions of China [25], India [42], Lao PDR [18,19,32], Malaysia [43], Philippines [29], Vietnam [30], Cambodia [31], Singapore [27], Nepal [44], and Pakistan [45], have characterized additional personal and community practices that can contribute to air pollution and poor lung health. Community practices such as trash, crop, and dung burning, household practices such as poor building material, overcrowding, exposure to dust and dirt, and occupational practices such as animal handling contribute to poor lung health [6]. These practices increase outdoor (ambient) air pollution and are becoming a growing concern for rural areas with growing urbanization [2,46]. Fine particles less than PM_10_ (particulate matter less than 10 microns) are considered most harmful because they are capable of deep lung penetration [47]. In high-income countries, it is rare for persons to exceed US-EPA 24-hour standard exposure to PM_10_. However, in rural dwellings in low resource settings, the standards are exceeded on a regular basis by a factor as high as 50--higher than outdoor levels found in coal-burning northern China [48]. 

The United Nations 2030 Sustainable Development Goals address air pollution and calls for a reduction in the mortality rate from the joint effects of household and ambient air pollution (Indicator 3.9.1) [49]. Achieving these goals will require the implementation of comprehensive multi-sectoral pollution control and risk mitigation interventions. A recent systematic review of the effectiveness of interventions for reducing household air pollution in homes continuing to use biomass fuels demonstrated improvements in indoor air quality (decreased average kitchen and personal levels of PM and CO). However, high statistical between study variability was observed, and post-intervention levels of pollutants continued to exceed WHO guidelines [50]. Therefore, current recommendations for the region continue to prioritize reducing the proportion of households using biomass fuel as the primary fuel [51]. The WHO has also recently launched a Clean Household Energy Solutions Toolkit (CHEST), which is designed to help countries develop contextualized policies and programs that expand clean energy access and use [52]. In regards to air pollution from crop fires, it is noted that while the intentional use of fires to transform land has declined globally, regulatory strategies have continued to be less effective in southern and eastern Asia. However, findings from a recent analysis of alternative crop residue burning approaches, conducted in India by Shyamsundar, et al. [53], strongly suggests that profitable and scalable alternatives exist and that India has an opportunity to be a model for other countries in the region [53,54]. 

## 5. Recommendations for Research

There is a clear opportunity to enhance global health outcomes through inter-sectoral commitment at the regional level to combat air pollution, biomass burning, and tobacco consumption. In 2019, the WHO declared air pollution as the greatest environmental risk to health [55]. The impact of atmospheric transport of pollution from distant sources has been well established, and cross-border air pollution is becoming an increasing source of tension between countries in Asia [56,57]. Despite having taken significant steps in recent years, China continues to be the largest emitter of CO_2_ emissions globally, due primarily to fuel combustion in energy production and burning of forests for agricultural needs [58]. In Lao PDR, there is a need to document and address the non-random spatial distribution of exposure at the China border. While Asian governments have in recent years signed several agreements to address cross-border air pollution, to prevent, mitigate and monitor wildfires and haze, and hold neighboring countries accountable, monitoring and enforcement mechanisms continue be fraught with challenges [56,58]. However, environmental experts report observing regional momentum and increasing political will to address transboundary pollution issues [56]. 

Within Lao PDR, there is a need for systems research to align agricultural, air pollution, and tobacco control initiatives to improve air quality and for systems research to be embedded in decision-making processes [59]. While health policy and health systems research tend to take the lead in these initiatives, there is a need to engage researchers and policymakers across multiple sectors. Additionally, there is a need for greater understanding of the practices and structures involved in successful inter-sectoral action, including the processes by which the obstacles to inter-sectoral action can be addressed [60]. The Health in All Policies initiatives where actors from multiple government sectors (e.g., social services, housing, transportation, education, employment relations, consumer protection, and environment) collaborate to address complex health problems provide promising exemplars, however, there continues to be a need for rigorous systematic research on their implementation to reveal successful practices [61].

Studies on air pollution exposure in Lao PDR are extremely limited. Studies have evaluated indoor air pollutant concentrations in the home, and their association with housing characteristics and respiratory symptoms (via questionnaire and peak expiratory flow rate measurements) among women and children [19,32,33], and associations between respiratory symptoms (via physical exam, survey and lung function: FEV1, FVC and FEV1/FVC ratio) and rates of exposure to multiple types of household and community level sources of air pollution among rural adult males [18]. However, there is a need for nationally representative studies that incorporate quantitative exposure data to evaluate the impact and association between air quality and respiratory disease in adults and children in Lao PDR. Additionally, according to Guinot and Annesi-Maesano [62], when compared with other regions of the world, differences in patterns of exposure and biological responses have been observed in Asia. Understanding these differences will require integrated research units to tackle the unique methodological and technical challenges represented by these differences. 

## 6. Limitations

Limitations of this study need mention. Our measure of household air pollution and other sources of smoke exposure within the home and in the community relied on self-report, and the survey items inquired about the potential for exposure to varied sources of pollution (vs. actual). However, the survey items did not capture differences within the types of exposures surveyed. For example, with indoor cooking fire smoke, the type of biomass fuel used for cooking was not assessed (e.g., coal vs. wood vs. crop residue). There are multiple factors that may affect the toxicity of chemicals and PM found in the varied sources of air pollution, including source, and composition of PM. Therefore, future studies should include monitoring of household and personal ambient exposures to particulate matter (PM_2.5_),carbon monoxide (CO), and biomarker data. 

## 7. Conclusions

Our findings from a national sample of adults in Lao PDR indicate that exposure to smoke from tobacco and indoor cooking fires was highly prevalent (over 74% for each exposure) and one-third of the households reported simultaneous exposure to smoke from tobacco and indoor cooking fires. When evaluated alongside our pilot studies indicating impaired lung function in the rural areas of Lao PDR [15], the high national prevalence of these harmful exposures underscores the need for tobacco control and NCD programs in the nation to focus on respiratory health risks in the homes. Initiatives focused on (1) replacing indoor cooking fires with clean-burning fuels, (2) crop burning alternatives and (3) tobacco control need a systems approach to enhance their efforts. Finally, due to the variability of household and community-level sources of air pollution, there is a need for enhanced surveillance and prioritization of at-risk regions and populations.

## Figures and Tables

**Figure 1 ijerph-16-03500-f001:**
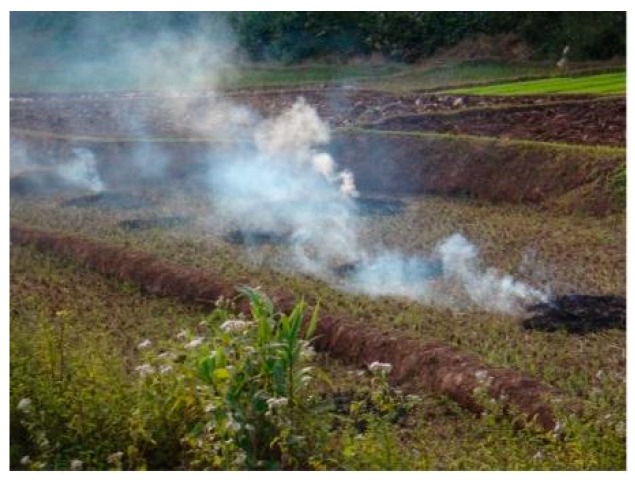
**Outdoor Crop Burning:** Example of the survey pictogram of rural life used to measure one source of air pollution.

**Table 1 ijerph-16-03500-t001:** Overall prevalence of exposure to household and community sources of air pollution (95% CI) among 9043 subjects of the 2012 National Adult Tobacco Survey of Laos (NATSL 2012) and prevalence by demographic variables [95% CI].

	Prevalence by Individual Household and Community Sources of Air Pollution				
	Exposure to SHS in the Home % [95% CI]	Exposure to Indoor Cooking Fires % [95% CI]	Exposure to Outdoor Cooking Fires % [95% CI]	Exposure to Trash Fires % [95% CI]	Exposure to Crop Fires % [95% CI]
**Overall**	74.5 [70.9,77.8]	78.0 [74.8,80.8]	56.0 [50.6,61.1]	38.6 [34.7,42.8]	30.1 [25.4,35.3]
**Age**					
≥15 to <25	73.9 [69.7,77.7]	75.6 [71.7,79.1]	52.4 [46.9,57.9]	34.0 [29.6,38.7]	27.3 [22.3,32.9]
≥25 to <45	74.4 [70.4,78.0]	79.8 [76.5,82.8]	57.8 [52.0,63.4]	38.7 [34.5,43.1]	30.6 [25.4,36.3]
≥45 to <65	75.3 [71.3,79.0]	79.8 [76.1,83.0]	58.3 [52.4,63.8]	45.5 [40.8,50.2]	33.0 [28.1,38.3]
≥65	76.6 [71.0,81.5]	73.4 [68.2,78.0]	55.0 [47.8,62.0]	41.8 [35.5,48.3]	33.2 [26.3,41.0]
**Gender**					
Male	75.1 [71.4,78.5]	73.3 [69.2,77.1]	53.4 [48.0,58.8]	40.0 [36.0,44.2]	30.8 [26.0,36.2]
Female	74.0 [70.3,77.3]	82.5 [79.8,85.0]	58.3 [53.0,63.5]	37.3 [33.2,41.6]	29.5 [24.8,34.6]
**Village Type**					
Urban	60.6 [53.0,68.0]	66.8 [60.8,72.2]	39.0 [30.6,48.1]	24.9 [18.1,33.2]	15.5 [10.6,22.0]
Rural, with Roads	80.0 [76.0,83.6]	83.4 [78.7,87.2]	63.4 [56.2,70.1]	48.7 [42.5,54.9]	37.8 [31.2,44.8]
Rural, without Roads	85.9 [72.0,93.5]	82.5 [67.4,91.5]	65.5 [44.5,81.8]	26.8 [16.8,39.8]	32.7 [13.7,60.0]
**Income**					
<1 USD/day	82.3 [73.8,88.4]	88.4 [81.5,93.0]	60.7 [48.6,71.7]	31.0 [24.5,38.5]	38.0 [24.3,53.9]
1–2 USD/day	78.1 [ 71.4,83.6]	76.3 [69.6,82.0]	58.0 [49.4,66.2]	30.1 [24.0,37.1]	23.9 [18.3,30.5]
2–3 USD/day	73.3 [66.9,78.8]	81.0 [76.1,85.1]	58.4 [50.7,65.8]	45.8 [38.9,52.7]	33.4 [26.6,41.0]
>3 USD/day	72.1 [67.8,76.0]	75.0 [71.3,78.2]	53.0 [46.7,59.2]	40.3 [35.1,45.7]	29.0 [24.0,34.4]
**Education**					
No Education	84.6 [80.2,88.2]	87.1 [83.0,90.4]	68.8 [60.0,76.5]	40.9 [34.2,48.1]	35.1 [24.2,47.8]
0–6 years	79.8 [76.0,83.2]	82.2 [78.5,85.4]	61.2 [55.4,66.7]	44.3 [39.4,49.3]	34.8 [29.6,40.4]
7–12 years	70.6 [66.2,74.7]	74.0 [70.1,77.6]	50.0 [44.1,56.0]	35.4 [30.6,40.7]	26.5 [22.0,31.6]
Vocational	66.5 [59.0,73.3]	68.2 [61.3,74.4]	46.8 [38.0,56.0]	33.0 [25.5,41.4]	22.4 [16.3,30.0]
University	47.8 [38.1, 57.6]	61.4 [54.7,67.7]	36.0 [26.2,46.1]	22.4 [15.5,31.2]	17.2 [11.7,24.5]

**Table 2 ijerph-16-03500-t002:** Overall prevalence of exposure to multiple combined sources of household and ambient air pollution among 9043 subjects of the 2012 National Adult Tobacco Survey of Laos (NATSL 2012) and prevalence by demographic variables (95% CI).

	Prevalence of Exposure to Multiple Sources of Air Pollution			
	Exposure to SHS + Indoor Cooking Fires % [95% CI]	Exposure to SHS + Indoor Cooking Fires + Outdoor Cooking Fires % [95% CI]	Exposure to SHS + Indoor Cooking Fires + Outdoor Cooking Fires + Trash Burning % [95% CI]	Exposure to SHS + Indoor Cooking Fires + Outdoor Cooking Fires + Trash Burning + Crop Burning % [95% CI]
**Overall**	32.8 [29.1,36.8]	20.5 [17.3,24.1]	12.3 [9.8,15.3]	6.7 [5.0,8.8]
**Age**				
≥15 to <25	31.1 [27.0,35.6]	18.2 [15.0,21.8]	11.0 [8.5,14.1]	6.0 [4.3,8.5]
≥25 to <45	35.0 [30.6,39.7]	22.3 [18.4,26.8]	12.8 [10.1,16.0]	7.0 [5.2,9.4]
≥45 to <65	33.2 [29.4,37.3]	22.3 [18.6,26.8]	14.6 [11.5,18.3]	7.3 [5.3,10.0]
≥65	26.4 [21.6,31.7]	15.6 [11.9,20.1]	9.3 [6.4,13.4]	6.0 [3.5,9.8]
**Gender**				
Male	29.6 [25.9,33.7]	18.8 [15.6,22.3]	12.0 [9.5,15.1]	6.8 [5.1,9.0]
Female	36.0 [32.0,40.0]	22.2 [18.7,26.0]	12.6 [10.1,15.5]	6.6 [5.0,8.7]
**Village Type**				
Urban	41.5 [34.8,48.5]	19.5 [14.3,26.0]	7.7 [4.9,12.0]	4.1 [2.3,7.4]
Rural, with Roads	30.4 [25.8,35.3]	21.9 [17.7,26.8]	16.4 [12.6,21.0]	8.9 [6.3,12.5]
Rural, without Roads	21.0 [9.2,40.8]	16.2 [6.8,33.7]	4.7 [1.9,11.2]	2.7 [0.8,8.5]
**Income**				
<1 USD/day	23.2 [16.2,32.2]	14.0 [8.8,21.7]	5.0 [2.9,8.6]	3.0 [1.6,5.6]
1–2 USD/day	28.4 [23.0,34.4]	17.3 [12.2,24.0]	10.9 [6.8,17.0]	2.6 [1.3,5.4]
2–3 USD/day	32.7 [27.0,38.9]	21.0 [15.8,27.3]	15.5 [10.8,21.8]	7.7 [4.5,12.9]
>3 USD/day	36.6 [32.2,41.1]	22.8 [19.1,26.8]	13.2 [10.5,16.4]	8.4 [6.3,11.2]
**Education**				
No Education	24.3 [18.7,30.9]	17.6 [13.1,23.2]	10.5 [7.5,14.6]	5.3 [3.5,8.0]
0–6 years	31.5 [26.8,36.6]	21.5 [17.6,26.0]	13.9 [10.7,17.9]	6.8 [5.1,9.1]
7–12 years	35.4 [31.5,39.5]	20.7 [17.2,24.6]	12.3 [9.4,15.9]	7.5 [5.3,10.4]
Vocational	39.6 [32.8,46.8]	22.0 [16.3,29.0]	11.2 [7.1,17.3]	6.7 [3.8,11.6]
University	41.4 [34.6,48.5]	19.1 [12.3,28.4]	7.47 [4.5,12.2]	4.4 [2.2,8.7]

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
