# Peer review of "Air Pollution in a Nationally Representative Sample: Findings from the National Adult Tobacco Survey of Lao PDR"

_ijerph, 2019, doi:10.3390/ijerph16183500_

Round 1
Reviewer 1 Report
Sentences on lines 79, and 80 are unfinshed
line 108 - seems to suggest that single households were included.
This was a simple descriptive paper of a National Survey in Laos, of exposures to air pollution (including all types). It is clearly written and a pleasure to read. The discussion was thoughtful and I agree with the ideas of considering all air pollution in a more systems approach.
There are a few minor issues that need correcting, and I also think that a better description of the exposure data collection would be useful, such as how were e.g: indoor cooking fires using cow dung or trash as fuel classified?
Please see errors/suggestions to improve clarity below:
Sentences on lines 79, and 80 are unfinished.
line 108 - seems to suggest that only single households were included. SUGGEST: "...inclusive of all households from all provinces that were private and either single or multiple members".
Figure 2 is actually a table and is unnecessary. It would be better to include in text (with the 95% CI's). Or you could add them to table 1, first line.
Table 1 also needs to % (95% CI) added to the heading of each column.
Figure 3 is also a table, pls add it to table 2. It could also be first line or more obviously under the column headings. And again this needs the % (95% CI) label added to each column.
There are also some errors in the rows in this table - i.e check how they lined up. Is it possible to increase the size of the title column so that the titles are only on 1 or 2 lines.
Please take the results (i.e %'s and ref to figures etc) out of discussion.
Line 288 is also unfinished. I would suggest that the authors could discuss limitation issues about data collection, in particular exposure assessment. I am unfamiliar with cooking practices in Laos, however how were indoor cooking using cow dung or crop residue recorded (as examples)? And other similar issues.
Font for the references is incorrect.
Author Response
Reviewer 1: Response in Bold Italics
Sentences on lines 79, and 80 are unfinshed
line 108 - seems to suggest that single households were included.
Both issues have been addressed. Please see below.
This was a simple descriptive paper of a National Survey in Laos, of exposures to air pollution (including all types). It is clearly written and a pleasure to read. The discussion was thoughtful and I agree with the ideas of considering all air pollution in a more systems approach.
There are a few minor issues that need correcting, and I also think that a better description of the exposure data collection would be useful, such as how were e.g: indoor cooking fires using cow dung or trash as fuel classified?
Please see errors/suggestions to improve clarity below:
Sentences on lines 79, and 80 are unfinished.
Sentence is now complete.
line 108 - seems to suggest that only single households were included. SUGGEST: "...inclusive of all households from all provinces that were private and either single or multiple members".
Revised as suggested.
Figure 2 is actually a table and is unnecessary. It would be better to include in text (with the 95% CI's). Or you could add them to table 1, first line.
Data has been added to Line 1 of Table 1 and Figure 2 deleted as suggested.
Table 1 also needs to % (95% CI) added to the heading of each column.
Thank you for catching these details. Has now been added to both Table 1 and 2.
Figure 3 is also a table, pls add it to table 2. It could also be first line or more obviously under the column headings. And again this needs the % (95% CI) label added to each column.
Revisions have been made as recommended. Data has been added to Table 2 and Figure 3 deleted.
There are also some errors in the rows in this table - i.e check how they lined up. Is it possible to increase the size of the title column so that the titles are only on 1 or 2 lines.
Edits have been made to table formatting throughout, and titles reduced to 3 lines.
Please take the results (i.e %'s and ref to figures etc) out of discussion.
Removed.
Line 288 is also unfinished. I would suggest that the authors could discuss limitation issues about data collection, in particular exposure assessment. I am unfamiliar with cooking practices in Laos, however how were indoor cooking using cow dung or crop residue recorded (as examples)? And other similar issues.
Limitations related to data collection have been expanded.
Font for the references is incorrect.
Reformatted and reviewed for accuracy.
Reviewer 2 Report
This study provides a general picture on air pollution exposure from different sources in Lao PDR using the prevalence of exposure, stratified by socioeconomic and demographical positions. In general, this study is informative, and the data/method are solid. My concern lies in the discussion section, limitation and recommendation section.
For discussion section, I would expect some discussion regarding the stratification and exposure level, but the discussion seems little bit detached from the result, more on health risks in general.
For limitation section, it is obvious, an unfinished sentence…
For the recommendation section, the author provides some recommendation for future research, but I would prefer(I'm sure the readers as well) more recommendations on pollution control policies or risk mitigation policies deriving from the result, which the author mentioned in the conclusion section, but not sufficient.
Author Response
Reviewer 2: Response in Bold Italics
This study provides a general picture on air pollution exposure from different sources in Lao PDR using the prevalence of exposure, stratified by socioeconomic and demographical positions. In general, this study is informative, and the data/method are solid. My concern lies in the discussion section, limitation and recommendation section.
For discussion section, I would expect some discussion regarding the stratification and exposure level, but the discussion seems little bit detached from the result, more on health risks in general.
We agree. Our findings that stratify on indoor/outdoor pollution exposures needed further discussion and we added that to the second and third paragraphs of the discussion.
For limitation section, it is obvious, an unfinished sentence…
Limitation section has been expanded.
For the recommendation section, the author provides some recommendation for future research, but I would prefer (I'm sure the readers as well) more recommendations on pollution control policies or risk mitigation policies deriving from the result, which the author mentioned in the conclusion section, but not sufficient.
Appreciate the suggestion. We have added references to control policies and risk mitigation strategies for tobacco, household air pollution and crop burning. Please see Lines 258-272.
Thank you for your careful review of the manuscript. We feel it has been strengthened by your suggestions.
Round 2
Reviewer 2 Report
accept for publication